# Generative and Contrastive Self-Supervised Learning for Virulence Factor Identification Based on Protein–Protein Interaction Networks

**DOI:** 10.3390/microorganisms13071635

**Published:** 2025-07-10

**Authors:** Yalin Yao, Hao Chen, Jianxin Wang, Yeru Wang

**Affiliations:** 1School of Information, Beijing Forestry University, Beijing 100083, China; yaoyalin@bjfu.edu.cn (Y.Y.); chenhaoxdy@outlook.com (H.C.); 2Risk Assessment Division 1, China National Center for Food Safety Risk Assessment, Beijing 100022, China

**Keywords:** virulence factor identification, protein–protein interaction, graph neural network, self-supervised learning

## Abstract

Virulence factors (VFs), produced by pathogens, facilitate pathogenic microorganisms to invade, colonize, and damage the host cells. Accurate VF identification advances pathogenic mechanism understanding and provides novel anti-virulence targets. Existing models primarily utilize protein sequence features while overlooking the systematic protein–protein interaction (PPI) information, despite pathogenesis typically resulting from coordinated protein–protein actions. Moreover, a severe imbalance exists between virulence and non-virulence proteins, which causes existing models trained on balanced datasets by sampling to fail in incorporating proteins’ inherent distributional characteristics, thus restricting generalization to real-world imbalanced data. To address these challenges, we propose a novel Generative and Contrastive self-supervised learning framework for Virulence Factor identification (GC-VF) that transforms VF identification into an imbalanced node classification task on graphs generated from PPI networks. The framework encompasses two core modules: the generative attribute reconstruction module learns attribute space representations via feature reconstruction, capturing intrinsic data patterns and reducing noise; the local contrastive learning module employs node-level contrastive learning to precisely capture local features and contextual information, avoiding global aggregation losses while ensuring node representations truly reflect inherent characteristics. Comprehensive benchmark experiments demonstrate that GC-VF outperforms baseline methods on naturally imbalanced datasets, exhibiting higher accuracy and stability, as well as providing a potential solution for accurate VF identification.

## 1. Introduction

Virulence factors (VFs) are essential for pathogenesis, enabling bacteria to adhere to and invade host cells, utilize host cell nutrients for survival and reproduction, evade host immune defenses, and secrete toxins that destroy host cells. Antibiotics remain the primary therapeutic agents for bacterial infections, exerting their effects by either killing bacteria or inhibiting their growth and reproduction. For example, this can be accomplished by disrupting the bacterial cell wall and cell membrane, inhibiting bacterial DNA/RNA and protein synthesis, and blocking folate synthesis [1]. Since antibiotics directly suppress bacterial survival, large-scale antibiotic use accelerates the evolution of bacterial antibiotic resistance. Antibiotic resistance has become a significant global public health challenge [2]. To relieve the pressure stemming from the rapid evolution of bacterial resistance, relevant research has proposed anti-virulence strategies aimed at achieving therapeutic effects by targeting VFs to suppress the manifestation of bacterial virulence. These strategies include inhibiting bacterial toxins’ gene expression and toxins transmission, disrupting bacterial adhesion capabilities, and interfering with bacterial communication [3]. Therefore, identifying bacterial VFs and developing an in-depth understanding of their mechanisms of action provide potential targets for the development of anti-virulence therapeutic strategies.

Due to the importance of this problem, a large number of studies have emerged in the field of VF identification. Methods based on sequence similarity aim to obtain homologous sequences corresponding to a given query and determine the type of the given query based on the type of homologous sequences. However, due to the diversity of VFs, which are involved in various pathogenic processes such as adhesion, invasion, and toxicity, and exhibit species-specificity and host-specificity, many VFs may show only insignificant similarity to known protein sequences [4]. Moreover, pathogens continuously adapt to the environment during the evolutionary process, and their VFs are prone to alterations. Therefore, traditional sequence alignment methods such as BLAST [5] have limited performance in identifying diverse VFs and VFs with distant evolutionary relationships.

To address this challenge, methods based on predefined protein features have been proposed. These methods extract features such as protein sequences, physicochemical properties, and evolutionary information, and combine them with traditional machine learning models and deep learning models for VF identification. For example, VirulentPred [6] takes amino acid composition (ACC), dipeptide composition (DPC), higher-order dipeptide composition, and position-specific scoring matrixes (PSSMs) as inputs for the first layer of a two-stage cascade support vector machine (SVM). The results from the first layer are then cascaded to the second-layer SVM classifier for training to generate the final classifier, which outperforms SVM classifiers based on single or multiple sequence features only in the first layer. MP3 [4] uses AAC and DPC as inputs for SVM, while leveraging the hidden Markov model (HMM) to analyze domain information in a local MiniPfam database constructed based on the Pfam database. The results from SVM and HMM are integrated according to specific rules, providing a new and promising approach for predicting virulence proteins in large-scale genomic or metagenomic datasets. PBVF [7] conducts relevant analysis and prediction of VFs by taking DPC and multiple sequence, similarity-based features as the inputs for SVM. Regarding the critical issue of negative dataset selection, this research adopts the NExIGO method based on Gene Ontology (GO) annotations to construct the negative dataset. The research results indicate that direct sequence similarity is crucial in the identification of VFs, and thus this characteristic should be fully utilized to improve the accuracy of analysis and prediction. DeepVF [8] extracts features based on sequence, physicochemical properties, and evolutionary information as inputs for four classic traditional machine learning models. It maps protein sequences to numerical values in alphabetical order, generating 10-dimensional features based on different window lengths for three deep learning models. By employing a stacking strategy, DeepVF effectively combines these baseline models, significantly enhancing model performance, as well as providing an important reference for classification problems in bioinformatics. VF-Pred [9] is based on sequence features and physicochemical properties and can successfully generate the sequence similarity features proposed in reference [7]. Additionally, it introduces the sequence alignment feature, Seg-Alignment, to capture the percentage of the best sequence alignment with the negative and positive datasets. These features are input into traditional machine learning models, and various ensemble methods including stacking, voting, and boosting are used together to enhance the classification performance. Experimental results show that the sequence alignment feature significantly improves the accuracy of the adopted machine learning algorithms. Methods based on predefined protein features usually rely on the knowledge of domain experts for design, and thus have certain limitations in feature selection, especially for manually extracted sequence features, which makes it difficult to comprehensively cover the potential information of protein sequences. Since the selection of predefined features is crucial to model performance, the model performance often highly depends on the quality of feature engineering.

With the rapid development of deep learning, natural language processing (NLP) has provided a new research perspective for protein representation learning [10,11]. Essentially, protein sequences can be viewed as a kind of “language”. By leveraging the framework of language models, it is possible not only to capture the local patterns within the sequences but also to reveal their global connections. Compared with traditional manually extracted features, protein features extracted based on language models are more likely to unearth the deep semantic information and complex patterns of protein sequences. The DTVF method proposed in reference [12] is based upon the protein sequence model ProtT5 [13], which is pre-trained on large-scale protein sequence data, as a protein feature extractor. It adopts a dual-channel architecture model, combining the long short-term memory network (LSTM) module with the convolutional neural network (CNN) module, and it introduces a dot-product self-attention layer into each respective module. This model can enhance the accuracy and efficiency of VF identification. Another approach, GTAE-VF [14], aided with a graph transformer autoencoder for VF identification, leverages the ESM-2 [15] language model to obtain amino acid feature vectors, and is the first to utilize three-dimensional structural information of protein predicted by ESMFold, finally transforming VF identification into a graph-level prediction task. The encoder–decoder framework integrates graph convolutional networks (GNNs) and Transformer structures, effectively capturing long-range correlations, thereby further improving predictive performance. Methods based on language models for protein features indicate that the introduction of high-quality features may help models better capture the relationship between VFs and sequence patterns. The transition from methods based on predefined features to those based on language model features marks a shift in VF identification models from traditional manual feature extraction to data-driven automated representation learning.

Most relevant studies primarily focus on the properties of proteins, including sequences, physicochemical properties, evolutionary information, and structural characteristics, to identify VFs. However, protein–protein interaction (PPI) network information has also been demonstrated to have potential in VF prediction. Reference [16] proposes a method to predict VFs based on the PPI network through the number, type, and interaction weight of neighbor nodes. Reference [17] further enriches the direct interaction neighbors of proteins into KEGG pathways, calculates the pathway enrichment scores, and uses the random forest model for prediction. However, these methods only utilize direct protein interactions and explicit features, such as the number and labels of neighbors, as well as pathway enrichment scores, which are limited to shallow information within the direct neighborhood. They lack comprehensive modeling of broader contextual and deeper information on PPI networks and do not fully leverage the biological features of proteins themselves, such as sequence and structure. Moreover, explicit features rely on known protein functional annotations, and the models may be restricted by incomplete annotation information, making it difficult to comprehensively capture the implicit properties of unannotated proteins. Since the PPI network can be naturally modeled and mined using a graph structure, GNNs can be taken into account to address the above limitations.

GNNs can integrate and simultaneously deal with protein sequence features and interaction information, as well as effectively aggregate node information across network layers through message-passing mechanisms to capture deeper, non-linear features [18]. Most GNNs are based on the homophily hypothesis [19,20,21], which essentially reflects the principle that “birds of a feather flock together” [22], meaning that interconnected nodes in a graph tend to belong to the same category. Under the homophily assumption, node representations will be smoothed through the aggregation process, with each node acquiring additional information from neighboring nodes that likely share the same label.

The natural distribution of proteins in bacterial systems exhibits an inherent imbalance between virulence and non-virulence proteins, with virulence proteins constituting a minority class. Traditional modeling approaches have attempted to address this imbalance through various sampling techniques. Under-sampling strategies are commonly employed for strain-specific datasets to reduce the number of non-virulence proteins moderately (such as maintaining a ratio of 1:5 between the size of the negative dataset and the positive dataset), thereby mitigating class imbalance effects. Balanced sampling is performed for multi-strain datasets by randomly selecting negative samples to match the number of positive samples to construct a balanced dataset. Additionally, repeated learning of virulence proteins is performed to enhance model performance. However, these sampling approaches may inadvertently compromise the inherent diversity of the training data, potentially limiting the model’s generalization capabilities. In imbalanced datasets, the majority class often dominates the learning process, causing the decision boundary to be biased towards the majority class. Self-supervised learning [23] has emerged as a promising alternative, offering the ability to extract effective representations from unlabeled data. This approach has the potential to capture more comprehensive and generalizable information that is inherent in the data itself, regardless of the skewed label distribution, demonstrating particular promise in addressing imbalanced classification challenges [24,25]. Existing self-supervised graph learning methodologies can be categorized into three distinct paradigms [26]. The generative approach leverages intrinsic graph information as self-supervised signals, focusing on reconstructing specific components of the input data to develop robust graph data representations [27,28,29]. Contrastive methods emphasize the analysis of consistency between different views to extract essential features and structural patterns inherent in the data [30,31]. The predictive approach autonomously generates informative labels as supervision signals, addressing the relationships between data and their corresponding labels [32,33].

In the field of class-imbalanced node classification, multiple innovative, self-supervised learning methods have been applied to meet the challenges. INS-GNN [34] implements generative, self-supervised pre-training to reconstruct the origin graph structure, effectively mitigating the inherent label bias present in imbalanced datasets. This method further incorporates self-training for pseudo-label assignment to unlabeled nodes and utilizes self-supervised edge enhancement to modify the structural characteristics of minority nodes, thereby amplifying their influence in the learning process. GraphMixup [35] improves the class-imbalanced node classification on graphs based on predictive self-supervised learning. It conducts semantic-level feature mixing by building a semantic relationship space and mixing edges while using an edge predictor trained via two context-based self-supervised tasks. Existing self-supervised learning approaches for addressing class-imbalanced graph node classification have been predominantly validated in conventional graph domains, such as social networks and citation networks, yet with limited exploration beyond these traditional contexts. This constraint is particularly noteworthy in the domain of bioinformatics, where complex graph structures such as PPI networks present unique challenges and opportunities that remain largely unexplored within the current methodological framework.

We present a novel framework for VF identification based on PPI networks, which integrates both generative and contrastive self-supervised learning strategies. Specifically, we construct local subgraphs centered on target proteins and generate multi-view representations through data augmentation strategies. These subgraph views are processed through GNNs to learn latent node representations. During the self-supervised learning phase, generative learning captures the inherent distributional characteristics of the data through node attribute reconstruction, while contrastive learning performs node-level comparisons between different views to effectively capture both local node features and contextual information. The learned latent representations are then utilized by a classifier for VF prediction.

The main objectives of this work are as follows:To develop a GNN-based framework for VF identification that leverages both the topological information of PPI networks and protein sequence features, transforming VF identification into a class-imbalanced node classification problem within the graph domain.To propose a novel self-supervised learning framework that combines generative and contrastive strategies, aiming to improve performance in imbalanced classification tasks through multi-view contrast and attribute reconstruction.

## 2. Materials and Methods

As illustrated in Figure 1, the Generative and Contrastive self-supervised learning framework for Virulence Factor identification (GC-VF) framework’s overall architecture comprises three components: protein sequence encoding, generative and contrastive self-supervised learning, and VF probability prediction. During the protein sequence encoding phase, the model encodes amino acid sequence features through multiple layers of deep neural networks, transforming them into protein sequence representations that are conducive to subsequent tasks. Subsequently, leveraging PPI networks, the model employs a hierarchical sampling strategy to construct local subgraphs of target proteins. Through data augmentation techniques, multi-view features essential for contrastive learning are generated. These views undergo information aggregation via GNNs, which enhance node representations by capturing more effective features.

In the self-supervised learning module, generative learning captures the intrinsic feature distribution of proteins by node attribute reconstruction tasks, while contrastive learning focuses on optimizing discriminative features between nodes by minimizing feature distances between positive samples and maximizing those between negative samples. Combining these approaches enables the model to take account of both global distribution and local variations in the data. Node-level generative and contrastive objectives inhibit decision-making from being dominated by global structural patterns, thus more precisely capturing the inherent characteristics of individual nodes. Finally, the model inputs the learned node representations into a classifier and utilizes the optimized features for VF prediction.

### 2.1. Datasets

Since the proportion of VFs in most bacterial strains is relatively low, three species with a moderate proportion of VFs were selected for analysis: *Salmonella enterica* serovar Typhimurium LT2, *Campylobacter jejuni* NCTC 11168, and *Staphylococcus aureus* NCTC 8325. Protein sequence data and PPI network data for these three strains were obtained from the STRING [36] database while corresponding VFs’ information was retrieved from the VFDB database [37]. Following the combined score threshold settings of the STRING database, only interaction relationships with a combined score greater than or equal to 0.4 were retained. To achieve a more comprehensive understanding of the characteristics inherent in each dataset, we calculated the basic properties of the PPI networks for these three strains and the class imbalance ratio (defined as the ratio of the number of samples in the largest class to that in the smallest class) [38] for each dataset. The relevant data is summarized in Table 1. Specifically, the numbers of VFs in the three strains are 156, 130, and 97, respectively. Furthermore, the class imbalance ratios in the selected datasets are 27.53, 11.48, and 31.72, respectively, all of which are notably high, indicating significant class imbalance. In this study, the class imbalance ratio is defined as the number of samples in the majority class divided by that in the minority class, a formulation that applies to binary classification settings.

### 2.2. Protein Sequence Encoding

Currently, significant advances have been achieved in extracting protein features from amino acid sequences through the application of protein language models [39,40]. In particular, the extraction method of amino acid embedding proposed in reference [41] has demonstrated substantial improvements in PPI-related tasks. This method implemented a dual-component embedding representation, the first leveraged amino acid sequences to compute co-occurrence similarities via a pre-trained Skip-Gram model [42], which was trained on the SHS148k dataset from the STRING database. Each protein sequence was treated as a sentence, and each amino acid as a token, allowing the model to learn contextual similarity by predicting surrounding amino acids (within a context window of 7) for each central amino acid, using negative sampling (size 5) to distinguish true co-occurrences from random noise. The second component employed one-hot encoding to capture electrostatic and hydrophobic similarities among amino acids [43]. Extending this amino acid embedding framework, reference [44] developed a multi-layer deep neural network architecture to generate protein feature representations optimized for PPI network input and achieved good predictive performance on PPI datasets using the same pre-trained embeddings from reference [41]. This architecture combined four types of layers, namely convolutional layers (Conv1d), pooling layers, bidirectional gated recurrent units (BiGRU), and fully connected (FC) layers to produce 256-dimensional protein representations. In our present work, we utilize this combination of the amino acid embedding method and multi-layer deep neural network architecture with an additional dropout layer to generate node feature representations that effectively characterize both global and local protein sequence properties. These refined feature embeddings establish a robust foundation for subsequent graph learning processes that incorporate PPI networks.

### 2.3. Generative and Contrastive Protein Representation Learning

#### 2.3.1. Graph View Establishment

Based upon the protein sequence encoding module, which is responsible for transforming protein sequences into node feature embeddings essential for GNNs, we employ a hierarchical neighborhood sampling method to construct local subgraphs [21] to further mine the structural information within PPI networks. Specifically, within each batch iteration, we initially select target proteins as center nodes from the PPI network through sampling without replacement. Next, we construct local structures through a two-layer neighborhood sampling approach, which involved randomly sampling $K_{1}$ nodes from first-order neighbors that have direct physical interactions with the center protein, and further sampling $K_{2}$ second-order neighbor nodes for each selected first-order neighbor, thereby constructing an original view centered on the target protein. The first-order neighbors capture direct interaction relationships, while the incorporation of the second-order neighbors facilitates the detection of potential indirect interaction patterns. Then, we introduce Gaussian-distributed random noise into the protein features of the original view, forming an augmented view. Subsequently, both the original and augmented views are processed as dual-stream inputs through the GNNs, enabling the aggregation of multi-scale neighborhood information and facilitating the learning of contextual protein representations within the PPI network.

#### 2.3.2. Generative Attribute Reconstruction

The basic principle of an autoencoder is to map the input data into a latent representation space through an encoder, followed by the reconstruction of the original input through a decoder. Within the context of graph data, this architecture not only facilitates the learning of latent node representations but also enables the capture of intrinsic association patterns between nodes through reconstruction tasks in an unsupervised manner.

For node attribute reconstruction, we implement a strategic approach by not employing target node anonymization (setting features to zero), instead enabling the original attributes of target nodes to participate directly in the information aggregation process. This decision stems from the consideration that virulence proteins do not manifest strong homophily in their interaction patterns. Therefore, the non-anonymized reconstruction approach facilitates the more precise capture of critical shared features between target nodes and their neighborhood nodes, while circumventing the introduction of irrelevant noise that might emerge from excessive reliance on neighborhood information.

The PPI network is formally defined as G=(V,E), where V denotes the set of protein nodes and E represents the set of interaction relationships. For each target node i∈V, views $G_{i}^{ori}$ and $G_{i}^{aug}$ are derived from sampling and feature augmentation procedures, respectively. Notably, the augmentation operation exclusively modifies node features while preserving structural consistency between the two views, yielding identical adjacency matrices.

The GNN architecture facilitates node representation updates through iterative neighbor information propagation, effectively mapping high-dimensional features onto a low-dimensional space. In the initial phase, the original subgraph $G_{i}^{ori}$ of target node i is processed through a multi-layer GNN encoder, which can be formally represented as(1)$$h_{i}^{\left(l\right)}={GNN}_{enc}\left(h_{i}^{\left(l-1\right)},\;A_{i}\right)$$
where $h_{i}^{\left(l\right)}$ represents the latent embedding of node i at the l-th layer of the GNNs. $A_{i}$ is the adjacency matrix of the subgraph sampled for node i. The ${GNN}_{enc}(\cdot )$ is a GNN encoder consisting of L layers. The multi-layer structure enables the gradual aggregation of node information from more distant neighbors and effectively captures the multi-level collaborative patterns among proteins. Each layer of the GNN is defined as follows:(2)$$h_{i}^{\left(l\right)}=UPDATE(h_{i}^{\left(l-1\right)}\;,AGGREGATE({\{h}_{j}^{\left(l-1\right)},\;for\;each\;j\in S(i)\}))$$
where S(i) represents the set of adjacent nodes in the subgraph sampled for node i. In particular, at the input stage, ${h_{i}^{\left(0\right)}=x}_{i}^{ori}$, represents the initial encoded feature of the protein sequence of the node.

The architecture allows for various GNN implementations, including Graph Convolutional Network (GCN) [19], Graph Attention Network (GAT) [20], and Graph Isomorphism Network (GIN) [45], each employing distinct neighbor aggregation strategies. In this study, we employ GraphSAGE [21] as the backbone network due to its robust generalization capabilities, achieved mainly by training a set of aggregator functions rather than individual node embeddings. The specific formulation of GraphSAGE is as follows:(3)$$h_{i}^{\left(l\right)}=\sigma (W\cdot CONCAT(h_{i}^{\left(l-1\right)},\;AGGREGATE({e_{ji}h}_{j}^{\left(l-1\right)},\;for\;each\;j\in S\left(i\right))))$$
where σ(·) is the ReLU [46] activation function, W represents the weight matrix, and $e_{ji}$ represents the weight of the edge between the node pair (j,i). The incorporation of edge weights enables the model to differentiate varying interaction strengths between protein pairs, reflecting their distinct collaborative importance in biological processes. The model firstly performs weighted aggregation of features from all neighboring nodes of node i in the subgraph and then concatenates the aggregated information with the feature of node i at the (l−1)-th layer to obtain the latent representation at the l-th layer.

The decoder transforms the encoder-generated node embeddings back into original node features, guiding the encoder toward more significant node representations while continuing to capture latent structural graph relationships. The decoder utilizes a multi-layer perceptron (MLP) with an architecture matching the encoder in terms of layer count and parameters, which can be denoted as(4)$${\hat{x}}_{i}^{ori}={MLP}_{dec}(h_{i}^{\left(L\right)})\;$$
where the MLP decoder implements two-layer architectures: ${MLP\left(h_{i}\right)=W}^{\left(2\right)}\sigma (W^{\left(1\right)}h_{i})$. Specifically, the decoder transforms the L-th layer node representation $h_{i}^{\left(L\right)}\;$ into a reconstructed original encoded feature ${\hat{x}}_{i}^{ori}$ through a series of non-linear transformations.

The generative self-supervised learning framework optimizes itself by minimizing the reconstruction error between decoder-generated and origin embeddings. The mean squared error (MSE) quantifies the reconstruction loss, which can be specifically written as(5)$$L_{gen}=\frac{1}{N}\sum_{i=1}^{N}\frac{1}{d}{∥\;x_{i}^{ori}-{\hat{x}}_{i}^{ori}∥}^{2}\;$$
where d denotes the dimension of the original encoded feature $x_{i}^{ori}$. Through minimization of reconstruction loss, the model learns to extract feature patterns shared between the target node and neighboring nodes during reconstruction.

#### 2.3.3. Multi-View Local–Local Contrasting

Contrastive learning, an unsupervised learning method, facilitates the exploration of structural information and node relationships to learn feature similarities and differences, offering novel approaches for identifying potential virulence proteins. However, virulence proteins within PPI networks, compared to their non-virulence counterparts, typically struggle to form stable subgraph features. This characteristic presents significant challenges for graph-level contrastive learning in extracting positive sample features from heavily imbalanced virulence protein subgraphs. Pooling subgraph features as positive samples in contrastive learning may disproportionately reflect non-virulence protein patterns, potentially compromising virulence protein feature learning. To address this challenge, we implement a node-level (local–local) contrastive scheme enabling independent optimization of individual node features.

Specifically, the encoded features $x_{i}^{ori}\;$ from original view and $x_{i}^{aug}\;$ from augmented view of the target node i are processed through a shared GNN encoder ${GNN}_{enc}\left(\cdot \right)$ to generate node embeddings $h_{i}^{ori}\;$ and $h_{i}^{aug}$. To enhance representation quality, we incorporate a projection head Projector(·), implementing a two-layer MLP that projects node features onto a new feature space. Within this space, positive sample pair features exhibit increased proximity while negative sample pairs maintain greater separation, thereby improving representation quality [47]. This process is formalized as(6)$$z_{i}^{ori}=Projector\;(h_{i}^{ori})\;$$(7)$$z_{i}^{aug}=Projector\;(h_{i}^{aug})$$

During node-level contrastive instance sampling, $z_{i}^{ori}$ from the original view serves as the anchor, while $z_{i}^{aug}$ from the augmented view serves as the positive sample. For negative sample selection, inspired by GRACE [48], we employ a mixed sampling strategy across and within views: node embeddings excluding node i in the original view are selected as negative samples to enhance intra-view relationship understanding, while nodes other than the positive sample in the augmented view are selected to improve inter-view discrimination capability.

The contrastive self-supervised learning objective simultaneously minimizes representation distance between similar samples while maximizing it between dissimilar samples. We employ InfoNCE [47] to compute contrastive loss, maximizing mutual information between random events, which is a practical and powerful tool for extracting shared data information. The InfoNCE loss can be formulated as(8)$$L_{con}=-\frac{1}{N}\sum_{i=1}^{N}log\frac{\exp\left(sim\left(z_{i}^{ori},z_{i}^{aug}\right)/\tau \right)}{\exp\left(sim\left(z_{i}^{ori},z_{i}^{aug}\right)/\tau \right)+{NSC}_{i}}\;$$
where $sim\left(h_{v},h_{u}\right)$ denotes the cosine similarity between the node features of v and u, calculated as $\frac{h_{v}h_{u}}{∥h_{v}∥∥h_{u}∥}$. The temperature coefficient τ modulates similarity distribution sharpness, with smaller values producing more concentrated distributions and larger values yielding smoother distributions. ${NSC}_{i}$ represents negative sample contribution to the loss function, computed as(9)$${\;NSC}_{i}=\sum_{j\in G_{i}^{ori},\;j\neq i}exp(sim\left(z_{i}^{ori},z_{j}^{ori}\right)/\tau +\sum_{k\in G_{i}^{aug},\;k\neq i}exp(sim\left(z_{i}^{ori},z_{k}^{aug}\right)/\tau$$
where the first term aggregates similarity contributions from original-view negative samples to the anchor node, while the second term computes augmented-view negative sample contributions. For a subgraph of size |V|, both terms use |V|−1 negative samples.

### 2.4. VF Prediction

The protein embeddings optimized through self-supervised modules achieve an integration of protein sequences, physicochemical properties, as well as contextual information derived from the PPI network. These embeddings are subsequently processed through a classifier to predict VFs. The classifier architecture incorporates a FC layer and a dropout layer, the latter of which effectively mitigates overfitting. The final output is transformed into a probability value via the sigmoid activation function. The classifier can be mathematically expressed as(10)$${\hat{y}}_{i}=\varphi (W\cdot Dropout(h_{i}^{ori})+b)$$
where $h_{i}^{ori}$ denotes the protein embedding of node i learned by the GNN, W represents the FC layer weight matrix, b is the bias term, and φ represents the sigmoid activation function.

To address class imbalance in VF prediction, we implement focal loss [49] for classification loss. The focal loss dynamically adjusts sample weights, diminishing easily classified sample contributions while amplifying difficult-to-classify sample importance, thereby enhancing minority class recognition capabilities. The focal loss calculation introduces an intermediate variable ${pt}_{i\;}$, which is a combination of the true label and the predicted probability, and its specific definition is as follows:(11)$$\;\;{pt}_{i}=y_{i}{\hat{y}}_{i}+(1-y_{i})(1-{\hat{y}}_{i})$$
where ${\hat{y}}_{i}$ represents the predicted probability, and $y_{i}$ is the true label. The complete form of the focal loss can be formulated as(12)$$L_{cls}=-\frac{1}{N}\sum_{i=1}^{N}\left[\lambda {\left(1-{pt}_{i}\right)}^{\gamma }log({pt}_{i})\right]$$
where λ represents the class weight coefficient balancing positive and negative sample contributions, and γ denotes the focusing parameter controlling attention distribution between easy and difficult samples.

The final loss function integrates three essential components: reconstruction loss $L_{gen\;}$, contrastive loss $L_{con}$, and classification loss $L_{cls}$. Self-supervised learning components are balanced by coefficients α and β as follows:(13)$$L=\alpha L_{gen}+\beta L_{con}+L_{cls}$$

The overall workflow of the proposed GC-VF framework is shown in Algorithm 1. Firstly, we sample a batch of protein nodes from the PPI network. For each target node, we generate its original view and augmented view, then input these two views into the shared GNN encoder to extract the embeddings of the target node. After that, we decode and restore the target node embeddings in the original view and calculate the reconstruction loss. Next, we input the target node embeddings from the two views into the projection head and project them onto a new embedding space. Subsequently, we adopt a mixed sampling strategy between and within views for node-level contrastive learning and calculate the contrastive loss. Finally, we combine the generation, contrastive, and classification tasks, train the model through joint optimization of multiple objectives, and perform predictions on proteins.

**Algorithm 1.** The key algorithm for the proposed GC-VF framework
$\mathrm{Input}:\;\mathrm{PPI}\;\mathrm{graph}\;G=\left(V,E\right)$, $\mathrm{Initial}\;\mathrm{protein}\;\mathrm{embeddings}\;x_{i}$,
Maximum number of training epochs T,
Batch size BOutput: VF prediction probability y^i1:for each training epoch t∈1,2,…,T do:2:Randomly divide the protein nodes V into batches of size B.3:  for each batch b=v1, v2,…, vB do:4:for each node vi in b do:5:   Randomly sample a second-order subgraph of vi as Siori, and   generate Siaug by adding Gaussian noise to the features.6:Compute the protein embeddings hiori  and
hiori from the   GNN encoder using the embeddings of both views, xiori and   xiaug, via Equation (3).7:Perform attribute reconstruction on hiori via Equation (4) to obtain   the reconstructed protein attribute x^iori.8:Calculate the reconstruction loss Lgen using Equation (5) between   xiori and x^iori.9:Project the GNN-encoded target node embeddings from the   Giori and Giaug by the projection head to obtain latent    embeddings via Equation (6) and Equation (7), respectively.10:Perform contrastive learning using ziori and ziaug to   compute contrastive loss Lcon via Equation (8).11:Calculate the classification loss Lcls for VF prediction via    Equation (12).12:Update the model parameters by backpropagating the total loss   L via Equation (13).13:**end for**14:**end for**15:**end for**16:Predict the virulence factor probability y^i for node vi via Equation (11).

### 2.5. Experimental Settings and Evaluation Metrics

In experimental implementation, a two-layer GraphSAGE was employed as the GNN backbone, utilizing LSTM as the aggregation function. Given that the protein sequence encoding module generated embeddings of 256 dimensions, we configured the GNN architecture with hidden dimensions of 128 and 64 for the first and second layers, respectively. The noise hyperparameter was configured at 0.1, and the InfoNCE loss temperature hyperparameter was set to 0.03. For the composite loss function, we assigned a coefficient of 0.4 to the generative loss. As for the contrastive loss, its coefficient varied depending on the dataset, which was set to either 0.6 or 0.2. The focal loss hyperparameters λ and γ were set to 0.25 and 2. The subgraph sampling was established at a size of (6, 6), resulting in a subgraph with the size of 37 nodes for each target node.

For performance evaluation, we employed two primary metrics: the Area Under the Receiver Operating Characteristic Curve (AUROC) and the Area Under the Precision–Recall Curve (AUPRC). The AUROC provides a comprehensive assessment of the model’s discriminative capability between positive (majority) and negative (minority) samples, while the AUPRC specifically emphasizes performance on positive samples. To ensure a thorough evaluation, we incorporated additional supplementary metrics: sensitivity, specificity, F1-score, and Matthews Correlation Coefficient (MCC). Sensitivity is specifically utilized to evaluate positive sample identification capability, while specificity measures its discriminative ability for negative samples. The F1-score adeptly balances the accuracy and recall evaluations of positive class prediction, and the MCC comprehensively contemplates various prediction outcomes, rendering them particularly appropriate for the evaluation of imbalanced datasets.

Our model was trained using the Adam optimizer, with a learning rate of 0.001 and a weight decay coefficient of 0.0001. The datasets were partitioned into training, validation, and test sets in a 6:2:2 ratio. Throughout the training process, a total of 131 epochs were carried out, and the model’s performance was scrutinized on the validation set every 10 epochs. The optimal model was then selected and archived. During the testing phase, the model was subjected to 100 trials on the test set, and the average of the results was adopted as the final performance indicator. To expedite and optimize the training process, the batch size was set to 300. The GC-VF model was executed on a NVIDIA GeForce RTX 3080 GPU.

To comprehensively evaluate method performance in VF prediction, we conducted experiments on three representative bacterial strain datasets: *S. enterica* serovar Typhimurium LT2, *C. jejuni* NCTC 11168, and *S. aureus* NCTC 8325. These datasets exhibit VF ratios of 3.5%, 8.0%, and 3.1%, respectively, reflecting the characteristic class imbalance encountered in real-world applications.

For comparative analysis, we selected four representative baseline methods:BLAST (2.16.0) [5]: relies on sequence similarity; constructs a local database from training sequences and predicts test sequences based on the most similar match.VirulentPred 2.0 [50]: applies classical machine learning; evaluated using default parameters provided by its web platform.DeepVF [8]: combines traditional machine learning and deep learning; tested using default configurations from the official server.DTVF [12]: adopts a deep learning framework; uses ProtT5 to encode protein sequences and applies a pre-trained predictor for classification.

## 3. Results and Discussion

### 3.1. Structural and Biological Pre-Analysis for Applying GNNs on PPI Networks

We have employed three commonly used homophily metrics—edge homophily [51,52], node homophily [53], and class homophily [21]—to evaluate the homophily of three PPI networks: *Salmonella enterica* serovar Typhimurium LT2, *Campylobacter jejuni* NCTC, and *Staphylococcus aureus* NCTC 8325. In addition to comparing with three homophily graphs commonly used in node classification tasks, we also compared the homophily of PPI networks with the randomly generated networks. We ensured that the randomly generated networks are of the same size as the corresponding PPI networks, in order to eliminate the influence of network size differences and focus precisely on the homophily aspect during the comparison process. For each PPI network, 30 random graphs were generated with the same number of nodes and edges, as well as identical label distributions. We employed the Configuration Model to preserve the degree distribution while randomizing edge connections. The Configuration Model algorithm represents each node’s degree as a corresponding number of “stubs” (half-edges), then randomly pairs all stubs to form complete edges while avoiding self-loops and multi-edges. This randomization process maintains both the class proportions and the original degree distribution, allowing us to isolate the effect of class-based connectivity patterns on homophily while controlling for degree heterogeneity. The three homophily metrics were calculated and subjected to *t*-tests (*p*-values all less than 0.05). The results are shown in Figure 2.

The original PPI networks exhibited higher node and edge homophily compared to commonly used homophily graphs for node classification tasks, although class homophily was relatively low. Overall, the homophily metrics of the original PPI networks were promising, suggesting that homophily-based GNNs are likely to effectively aggregate node information to obtain beneficial embeddings. Compared to random networks of the same scale, the original PPI networks showed superior performance across all homophily metrics. The high node and edge homophily observed in both original PPI networks and random networks of the same size may reflect the impact of class imbalance in PPI networks, where dominant class nodes enhance these two metrics. The numerical advantage of dominant class nodes increases the probability of connections between nodes within this class. The difference in class homophily between original PPI networks and random networks indicates significant collaborative relationships among minority class nodes (virulence proteins), suggesting their coordinated participation in pathogenesis-related biological processes. Furthermore, based on the STRING database, KEGG pathway enrichment analysis was performed on the interactions between virulence proteins, and the average local clustering coefficient among virulence proteins in these pathways was calculated, with results shown in Figure 3. The high average local clustering coefficient indicates that virulence proteins cooperatively participate in pathogenesis-related biological activities through tight interactions, involving multiple aspects such as bacterial secretion regulation, bacterial invasion of host cells, synthesis and assembly of bacterial structural components, and coordination of quorum sensing. The intimate interactions of virulence proteins in PPI networks facilitate learning virulence protein features through message passing.

### 3.2. Baseline Methods Comparison

In performance evaluation, we employ multiple complementary metrics. The AUROC baseline of 0.5 represents random classifier performance, while AUPRC baselines vary with positive sample proportions, corresponding to 0.035, 0.08, and 0.031 for the three datasets, respectively. The experimental results presented in Table 2 demonstrate that our proposed GC-VF method exhibited notable overall performance.

Analysis of experimental results revealed that while certain baseline methods achieved high accuracy on specific datasets, such apparent advantages may mask underlying classification limitations. These methods often exhibited high specificity, showing a tendency to classify samples into the majority negative class. In scenarios with substantial class imbalance, while this tendency might yield favorable accuracy and specificity metrics, the actual discriminative capability remains limited. Thus, comprehensive evaluation necessitates consideration of multiple metrics, particularly sensitivity and AUPRC.

Notably, we observed that some methods often struggled to maintain adequate specificity while attempting to enhance sensitivity. For instance, VirulentPred 2.0, despite demonstrating high sensitivity, achieved a low F1-score, indicating that increased recall came at the cost of precision, resulting in numerous false positive predictions. This observation underscores the importance of balancing various performance metrics in VF prediction and highlights the GC-VF framework’s capability in effectively harmonizing these competing objectives.

### 3.3. Analysis of Graph-Based Approaches

We conducted a comprehensive evaluation of various graph methods on three bacterial strain datasets, categorized into two primary classes: graph learning and graph construction. To ensure experimental validity, consistent datasets and hyperparameter settings were maintained across all methods.

Among the graph learning methods, we primarily evaluated three prominent GNN architectures: GraphSAGE, GCN, and GAT. GraphSAGE aims to train aggregator functions and has multiple aggregator architectures, including the Mean aggregator, LSTM aggregator, and Pooling aggregator. Moreover, GraphSAGE-GCN, a convolutional variant of GraphSAGE, is an extended inductive version of GCN.

In terms of graph construction methods, we explored several graph construction strategies: the PPI networks, the Cosine Similarity Graph (CSG), and the BLAST Similarity Graph (BSG). The CSG model computed the cosine similarity between protein feature vectors and constructed the graph using a threshold of 0.5. The BSG model, on the other hand, calculated protein sequence similarities via BLAST (2.16.0) and retained edges with an E-value below 20. In addition, we set a baseline method without using a graph structure (MLP-NS), which directly inputs the embeddings encoded from protein sequences into an MLP classifier. Notably, both the CSG and BSG models utilized GraphSAGE-LSTM for graph learning.

Figure 4a,b present comparative AUROC and AUPRC metrics across the three bacterial strain datasets. In the experiments’ PPI of the graph-based GNN variant, GraphSAGE models consistently outperformed GCN and GAT, with superior performance from LSTM and Pooling aggregators. The LSTM aggregator achieved optimal performance, primarily due to its distinctive sequence processing mechanism. By establishing continuous dependency chains between nodes, it effectively captured complex neighbor relationships, which might have enhanced the GNNs’ expressive capacity in neighbor feature aggregation, enabling more comprehensive node-dependency modeling. Meanwhile, the Pooling aggregator, by adaptively identifying key features from neighbors, focused attention on important information for VF prediction.

Experimental results revealed significant performance degradation when excluding the PPI network structure. While both CSG and BSG approaches based on sequence similarity captured certain protein relationships, they failed to adequately capture complex biological interactions. This characteristic underscores the PPI network’s advantage in capturing protein functional relationships: sequence similarity alone does not equate to functional similarity. For instance, proteins with similar sequences may not participate in identical biological pathways, whereas PPI graphs better reflect proteins’ synergistic effects through real biological interactions. Furthermore, even advanced sequence-based models (e.g., CNN, GRU) underperformed when employing graph-based methods. This suggests that graph structures have more potential to capture protein feature relationships, compensating for the limitations of sequence-only models.

### 3.4. Hyperparameter Study

We conducted comprehensive experiments to evaluate how various key hyperparameters influenced our proposed framework’s performance. The analysis focused on four critical hyperparameters: noise hyperparameter, temperature hyperparameter, subgraph sampling size, and loss function balance factors. Below we present our detailed findings.

In our contrastive learning setup, we generated an augmented view by applying Gaussian noise to the original node features. The noise, following a standard normal distribution, was controlled by a hyperparameter that determined the perturbation magnitude. We tested noise hyperparameter values throughout the set {0.001, 0.005, 0.01, 0.03, 0.05, 0.07, 0.1, 0.3, 0.5, 0.7} and evaluated performances using AUROC and AUPRC, as illustrated in Figure 5a,b. Our findings revealed that moderate noise levels improved model performance, though the effect varied from dataset to dataset. Excessive noise (above 0.1) significantly degraded performance by disrupting feature integrity. Through extensive testing, we determined that a noise hyperparameter of 0.1 provided an optimal balance between data diversity and stability across datasets.

The temperature hyperparameter τ in the InfoNCE loss governs the model’s ability to distinguish between positive and negative samples. Our experiments, spanning τ values within the set {0.03, 0.05, 0.07, 0.09, 0.1, 0.3, 0.5, 0.7, 1.0}, revealed dataset-specific sensitivities to this hyperparameter (Figure 5c,d). When setting τ = 0.01, an abnormal situation occurred where the model outputs NaN (Not a Number) values. This indicated that the selected value for this parameter was rather diminutive, failing to support stable operations of the model. Consequently, in subsequent experiments regarding the setting of this hyperparameter, we initiated the range from 0.03 to avoid such instabilities and further explored the optimal configuration. Based on extensive testing, we finally selected τ = 0.03 as the optimal value, delivering consistent and superior performance across most datasets.

We implemented a two-layer subgraph sampling strategy, denoted as $\left(K_{1},\;K_{2}\right)$, where $K_{1},\;K_{2}\;\in$ {2, 3, 4, 5, 6, 7} and $K_{2}\;\leq \;K_{1}$. Results shown in Figure 6 indicate that subgraph size particularly affects AUPRC, while AUROC remains more stable. Larger first-layer sampling ($K_{1}$) generally enhanced performance, suggesting better capture of local information characteristics. Similarly, increased second-layer sampling ($K_{2}$) improved AUPRC, highlighting the importance of second-order neighborhood information in modeling contextual relationships. Balancing performance and computational efficiency, we established $\left(K_{1},\;K_{2}\right)$ = (6, 6) as optimal sampling hyperparameters.

Finally, we explored how the coefficients α and β in loss function (13) affected model performance by modulating the influence of generative and contrastive self-supervised modules. Given that self-supervised learning serves an auxiliary role to classification, we constrained both α and β to (0, 1]. We set the coefficients in a range of {0.2, 0.4, 0.6, 0.8, 1.0} and conducted a comprehensive exploration of different combinations within this range. Results in Figure 7 show that small coefficients (α = 0.2, β = 0.2) led to suboptimal performance due to insufficient feature learning, while large values (α = 1.0, β = 1.0) compromised classification capability by overemphasizing self-supervised tasks. The *C. jejuni* NCTC 11168 dataset showed particular sensitivity to the generative loss coefficient α, performing best with high α and low β values (β = 0.2). Conversely, the *S. enterica* serovar Typhimurium LT2 dataset exhibited performance degradation at α = 1.0, suggesting dataset-specific dependencies on different self-supervised learning modules. Based on these findings, we standardized α at 0.4 while allowing β to vary by dataset: β = 0.2 for *S. aureus* NCTC 8325 and β = 0.6 for the other datasets, achieving optimal cross-dataset performance.

### 3.5. Ablation Study

To systematically evaluate the GC-VF framework’s key components, we conducted a series of ablation experiments. We tested several variant models by removing specific modules: the generative attribute reconstruction module (GC-VF w/o Gen), the local contrastive learning module (GC-VF w/o Con), and the self-supervised learning module (GC-VF w/o SL). Additionally, we evaluated variants of the GraphSAGE encoder without edge weights (GC-VF w/o EW) and with binary cross-entropy loss replacing focal loss (GC-VF w/BCE). Table 3 presents the comparative results across all three datasets.

Our experiments revealed that the complete GC-VF model, incorporating both generative and contrastive learning for self-supervision, achieved notable performance across all datasets. The ablation studies demonstrated that removing any self-supervised module results in performance degradation. Notably, the combined implementation of both modules produced performance improvements exceeding their individual contributions, confirming positive and effective synergy between these self-supervised strategies. However, we observed distinct variations in module contributions across different datasets. For the *S. enterica* serovar Typhimurium LT2 and *C. jejuni* NCTC 11168 datasets, the removal of the generative module resulted in minimal impact, suggesting contrastive learning’s dominance in feature extraction. Conversely, for the *S. aureus* NCTC 8325 dataset, the attempt to remove the generative module significantly degraded performance across multiple metrics, underlining its crucial role in feature learning. These variations suggest that bacterial strain datasets possess unique feature distribution patterns, leading to differential responses to various self-supervised learning strategies.

The analysis of edge weights revealed their importance in model performance. Removing edge weights substantially degraded the results, primarily because traditional feature aggregation methods fail to capture the nuanced intensity differences in PPI networks. Our statistical analysis of PPIs across the three datasets revealed distinctive patterns in interaction strengths: virulence protein interactions averaged a combined score of 0.7556, virulence-to-non-virulence protein interactions averaged 0.6037, and non-virulence protein interactions averaged 0.6428. These findings indicate stronger interaction patterns between proteins of the same type, among which virulence proteins exhibit the strongest interaction patterns. This biological insight justifies our integration of edge weights into the framework. By incorporating these differentiated edge weights into the GNNs, our model effectively prioritizes strong interaction connections, particularly the high-intensity interactions between virulence proteins, during feature aggregation. This approach not only enhances the model’s prediction capabilities but also aligns with established biological characteristics of PPI networks.

Furthermore, the implementation of focal loss proved effective in addressing class imbalance, even with default hyperparameter settings. The comparative analysis showed that replacing focal loss with binary cross-entropy loss resulted in performance degradation, highlighting focal loss’s beneficial role in managing data imbalance challenges.

### 3.6. Training and Inference Efficiency

Training and inference runtimes are reported in Table 4. The runtime measurements represent averaged results using optimal hyperparameters for each dataset and were executed on a NVIDIA GeForce RTX 3080 GPU. As expected, both training and inference times correlate positively with graph scale, as larger networks require more computational resources. Training time includes the complete 131-epoch process using the Adam optimizer (learning rate of 0.001, weight decay of 0.0001) and a batch size of 300, while inference time refers to the average duration of 100 test evaluations on the respective test sets.

## 4. Conclusions

Experimental results on multiple real-world PPI network datasets demonstrated the effectiveness and robust performance of the GC-VF framework we have proposed. The key contributions can be summarized as follows:We employed GNNs to identify VFs leveraging PPI networks. This approach integrated both topological information from PPI networks and protein sequence features. Notably, we pioneered in transforming the VF identification task into a class-imbalanced node classification problem within the graph domain.We proposed a novel framework for VF identification that combines generative and contrastive self-supervised learning. Through attribute reconstruction and multi-view contrast, these two approaches worked synergistically to enhance model performance in imbalanced classification tasks.

Our approach fundamentally shifts virulence factor prediction from isolated sequence analysis to systems-level understanding by incorporating protein–protein interaction networks, reflecting the biological reality that virulence emerges from coordinated molecular interactions rather than individual proteins. By integrating self-supervised learning within a graph framework, GC-VF captures both intrinsic features and contextual dependencies while addressing class imbalance challenges commonly encountered in biological datasets. This work establishes a paradigm for modeling complex molecular behaviors through network-based representation learning, offering a generalizable framework that extends beyond virulence prediction to broader functional inference tasks in systems biology. In future work, we plan to enhance model transferability across diverse PPI networks through domain-adaptive strategies and explore automated hyperparameter optimization to improve efficiency. Additionally, we aim to extend this approach to host–pathogen interaction studies, integrating pathogen and human proteins into a unified PPI network to uncover critical host factors and novel therapeutic targets.

## Figures and Tables

**Figure 1 microorganisms-13-01635-f001:**
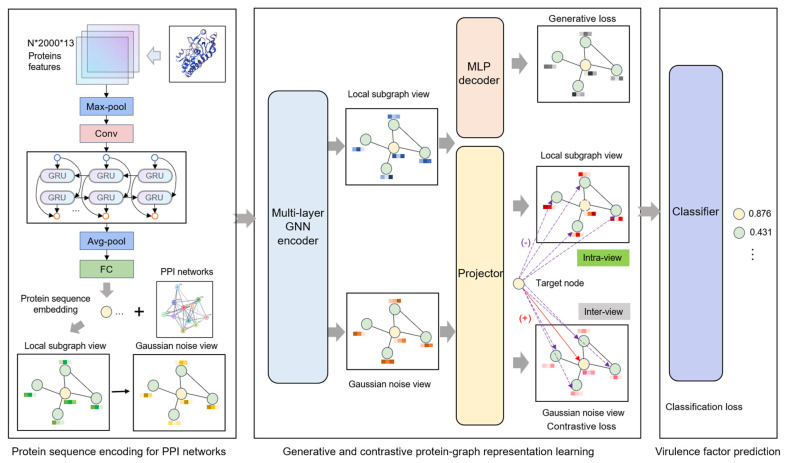
The overall framework of the Generative and Contrastive self-supervised learning framework for Virulence Factor identification (GC-VF).

**Figure 2 microorganisms-13-01635-f002:**
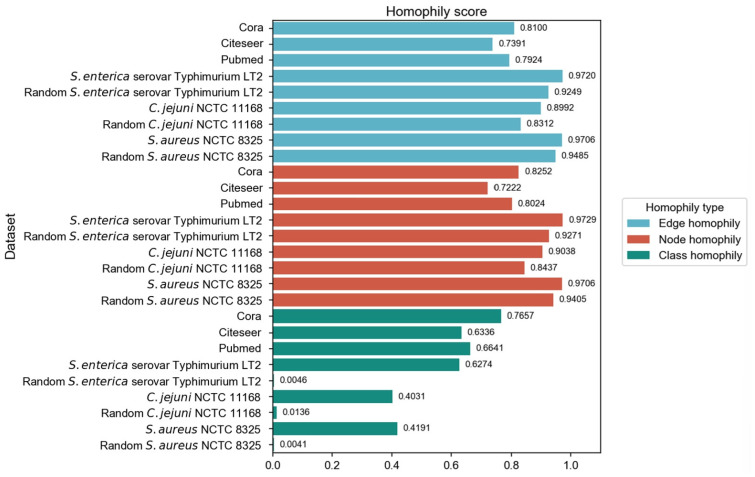
Comparison of homophily evaluation results across different graph types.

**Figure 3 microorganisms-13-01635-f003:**
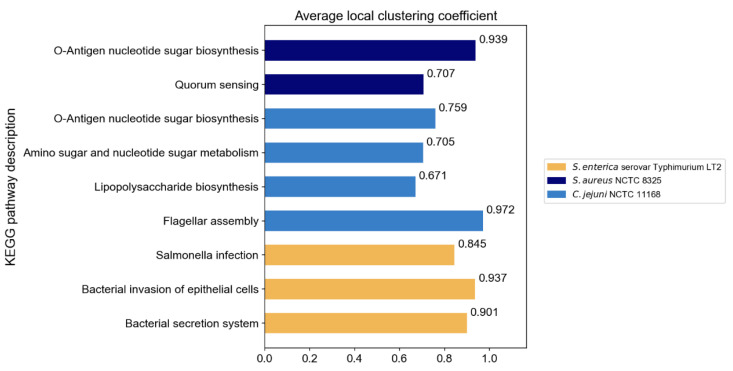
The average local clustering coefficient of virulence proteins in KEGG pathways.

**Figure 4 microorganisms-13-01635-f004:**
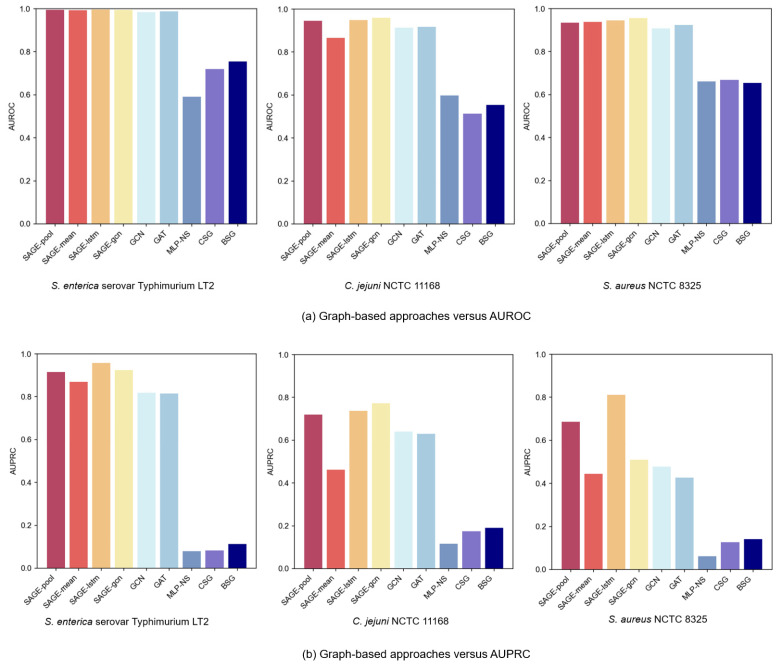
Performance comparison of graph learning and construction approaches.

**Figure 5 microorganisms-13-01635-f005:**
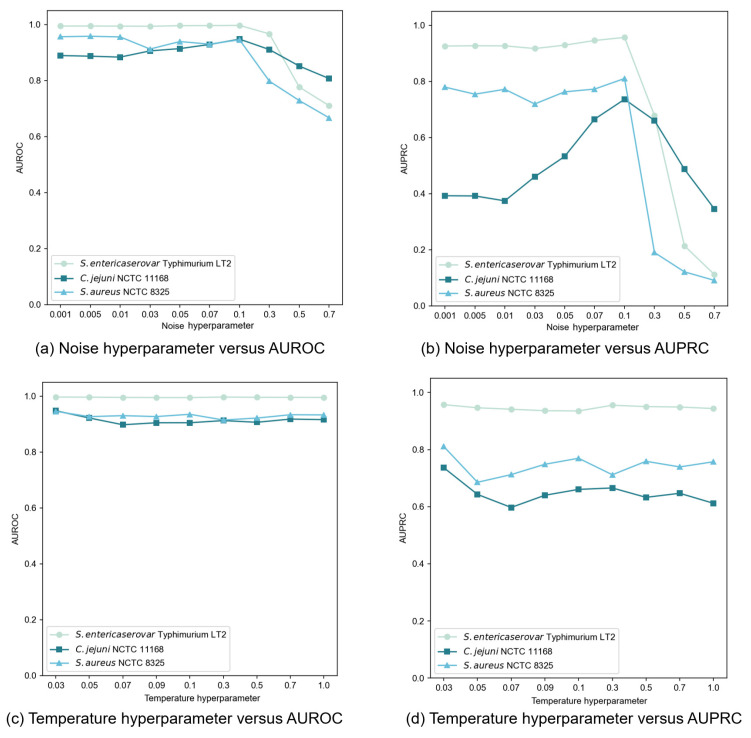
Sensitivity analysis of noise and temperature hyperparameters on GC-VF performance.

**Figure 6 microorganisms-13-01635-f006:**
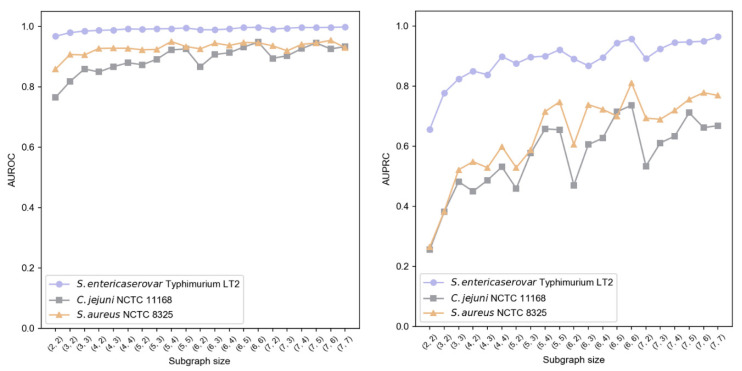
Impact of subgraph sampling size on GC-VF performance.

**Figure 7 microorganisms-13-01635-f007:**
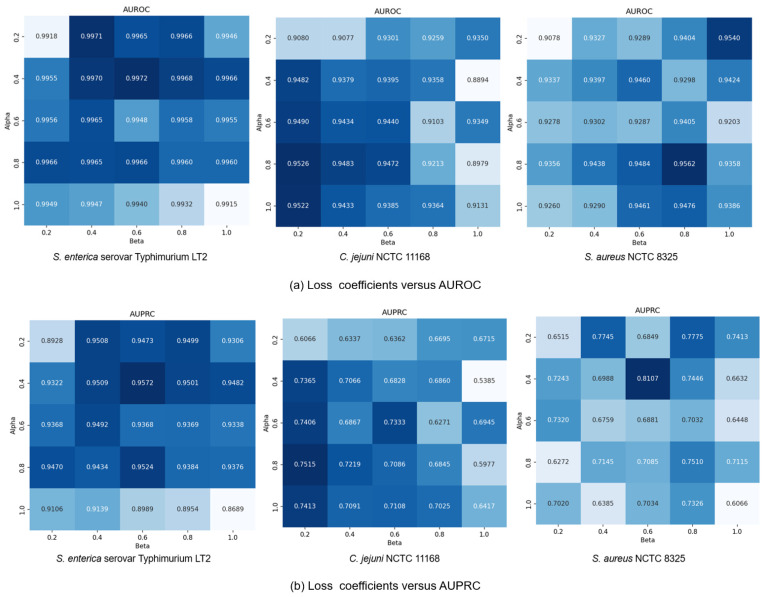
GC-VF performance variation with different α  and β combinations.

**Table 1 microorganisms-13-01635-t001:** Dataset statistics.

Dataset	Nodes	Edges	VFs	Imbalance Ratio
*S. enterica* serovar Typhimurium LT2	4451	86,605	156	27.53
*C. jejuni* NCTC 11168	1623	81,710	130	11.48
*S. aureus* NCTC 8325	2847	79,578	87	31.72

**Table 2 microorganisms-13-01635-t002:** Performance comparison of different methods for VF prediction.

Dataset	Method	Accuracy	Sensitivity	Specificity	F1-Score	MCC	AUPRC	AUROC
*S. enterica* serovar Typhimurium LT2	BLAST	0.9560	0.4839	0.9731	0.4348	0.4145	0.3440	
	VirulentPred 2.0	0.6979	**0.9677**	0.6881	0.1829	0.2552	0.0989	
	DeepVF	0.9008	0.3548	0.9208	0.2018	0.1788	0.2593	0.7942
	DT-VF	0.9651	0.3548	0.9871	0.4151	0.4038	0.2761	0.8678
	GC-VF	**0.9941**	0.9058	**0.9973**	**0.9140**	**0.9119**	**0.9572**	**0.9972**
*C. jejuni* NCTC 11168	BLAST	0.8738	0.3462	0.9197	0.3051	0.2388	0.3356	0.8738
	VirulentPred 2.0	0.4831	**0.9231**	0.4448	0.2222	0.2025	0.1228	
	DeepVF	0.6440	0.7308	0.6364	0.2484	0.2045	0.4510	0.6695
	DT-VF	0.6800	0.8077	0.6689	0.2877	0.2679	0.1817	0.7622
	GC-VF	**0.9588**	0.6877	**0.9824**	**0.7276**	**0.7107**	**0.7365**	**0.9482**
*S. aureus* NCTC 8325	BLAST	0.9579	0.3529	0.9765	0.3333	0.3122	0.3440	
	VirulentPred 2.0	0.4614	**1.0000**	0.4448	0.0997	0.1528	0.0525	
	DeepVF	0.5554	0.8750	0.5456	0.1041	0.1427	0.4670	0.7021
	DT-VF	0.9035	0.4706	0.9168	0.2254	0.2250	0.1678	0.8171
	GC-VF	**0.9889**	0.7524	**0.9961**	**0.8005**	**0.7990**	**0.8107**	**0.9460**

Bold values indicate the best results, and underlined values indicate the second-best results.

**Table 3 microorganisms-13-01635-t003:** Ablation experiment results of GC-VF key modules across three datasets.

Dataset	Model	Accuracy	Sensitivity	Specificity	F1-Score	MCC	AUPRC	AUROC
*S. enterica* serovar Typhimurium LT2	GC-VF w/o Con	0.9906	0.8458	0.9959	0.8625	0.8596	0.9106	0.9929
	GC-VF w/o Gen	0.9922	0.8632	0.9969	0.8855	0.8831	0.9151	0.9889
	GC-VF w/o SL	0.9894	0.8055	0.9961	0.8414	0.8380	0.8754	0.9813
	GC-VF w/o EW	0.9873	0.8084	0.9938	0.8168	0.8132	0.8689	0.9915
	GC-VF w/BCE	0.9775	0.7100	0.9872	0.6886	0.6823	0.7333	0.9824
*C. jejuni* NCTC 11168	GC-VF	**0.9941**	**0.9058**	**0.9973**	**0.9140**	**0.9119**	**0.9572**	**0.9972**
	GC-VF w/o Con	0.8803	0.6046	0.9042	0.4482	0.4087	0.3591	0.8688
	GC-VF w/o Gen	0.9447	0.5323	0.9806	0.6070	0.5928	0.5921	0.8748
	GC-VF w/o SL	0.9287	0.4519	0.9701	0.5065	0.4937	0.4833	0.8323
	GC-VF w/o EW	0.9325	**0.7012**	0.9526	0.6249	0.6000	0.6013	0.9365
*S. aureus* NCTC 8325	GC-VF w/BCE	0.9268	0.4646	0.9670	0.5061	0.4873	**0.8428**	0.4646
	GC-VF	**0.9588**	0.6877	**0.9824**	**0.7276**	**0.7107**	0.7365	**0.9482**
	GC-VF w/o Con	0.9856	0.6612	0.9955	0.7327	0.7401	0.7457	**0.9535**
	GC-VF w/o Gen	0.9806	0.7382	0.9880	0.6942	0.6877	0.6613	0.9296
	GC-VF w/o SL	0.9827	0.6835	0.9919	0.7035	0.7038	0.7301	0.9379
	GC-VF w/o EW	0.9841	0.6759	0.9936	0.7172	0.7141	0.6978	0.9342
	GC-VF w/BCE	0.9867	0.7124	0.9951	0.7610	0.7585	0.7538	0.9013
	GC-VF	**0.9889**	**0.7524**	**0.9961**	**0.8005**	**0.7990**	**0.8107**	0.9460

Bold values indicate the best results, and underlined values indicate the second-best results.

**Table 4 microorganisms-13-01635-t004:** Runtime performance.

Dataset	Nodes	Training Time (s)	Inference Time (s)
*S. enterica* serovar Typhimurium LT2	4451	282.42	65.00
*C. jejuni* NCTC 11168	1623	87.64	32.59
*S. aureus* NCTC 8325	2847	160.42	42.95

## Data Availability

The source code of this work can be found at https://github.com/Aysiling/GC-VF (available from 24 February 2025). The virulence factor data for *S. enterica* serovar Typhimurium LT2 were obtained from the VFDB at https://www.mgc.ac.cn/cgi-bin/VFs/compvfs.cgi?Genus=Salmonella (accessed on 24 October 2024). The virulence factor data for *C. jejuni* NCTC 11168 were obtained from VFDB at https://www.mgc.ac.cn/cgi-bin/VFs/compvfs.cgi?Genus=Campylobacter (accessed on 24 October 2024). The virulence factor data for *S. aureus* NCTC 8325 were obtained from VFDB at https://www.mgc.ac.cn/cgi-bin/VFs/compvfs.cgi?Genus=Staphylococcus (accessed on 24 October 2024). The protein sequence data for *S. enterica* serovar Typhimurium LT2 were downloaded from the STRING database at https://stringdb-downloads.org/download/protein.sequences.v12.0/99287.protein.sequences.v12.0.fa.gz (accessed on 24 October 2024). The protein sequence data for *C. jejuni* NCTC 11168 were obtained from https://stringdb-downloads.org/download/protein.sequences.v12.0/192222.protein.sequences.v12.0.fa.gz (accessed on 24 October 2024). The protein sequence data for *S. aureus* NCTC 8325 were obtained from https://stringdb-downloads.org/download/protein.sequences.v12.0/93061.protein.sequences.v12.0.fa.gz (accessed on 24 October 2024). The PPI data for *S. enterica* LT2 were retrieved from https://stringdb-downloads.org/download/protein.links.v12.0/99287.protein.links.v12.0.txt.gz (accessed on 24 October 2024). The PPI data for *C. jejuni* NCTC 11168 were obtained from https://stringdb-downloads.org/download/protein.links.v12.0/192222.protein.links.v12.0.txt.gz (accessed on 24 October 2024). The PPI data for *S. aureus* NCTC 8325 were obtained from https://stringdb-downloads.org/download/protein.links.v12.0/93061.protein.links.v12.0.txt.gz (accessed on 24 October 2024).

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
