# Peer review of "Generative and Contrastive Self-Supervised Learning for Virulence Factor Identification Based on Protein–Protein Interaction Networks"

_microorganisms, 2025, doi:10.3390/microorganisms13071635_

Round 1

Reviewer 1 Report

Comments and Suggestions for Authors

Introduction

The introduction should be revised to make it more concise and objective. Review the need to present figures 1 and 2 in the introduction as results of a methodological pre-analysis. Results not previously described or new approaches should be presented in the results section;

Line 120 -> Add spacing after "networks";

Line 152 -> "We employ" -> We have employed -> review English;

Line 157 -> "we also compare" -> we have also compared -> review English;

Line 161 -> "are generated" -> were generated -> review English;

Line 162 -> "are calculated and subject to" -> were calculated and subjected to -> review English;

Line 246-258 -> "The key contributions can be summarized..." -> this section should be revised to include the objectives of the work; it is suggested that "contributions" be moved to the conclusions section.

Materials and Methods

Line 260 -> GC-VF -> Introduce GC-VF abbreviation in the main text and in the caption of figure 3, as it was only introduced in the abstract;

Figure 3 -> Legend -> "Virulence factor predication" -> Virulence factor prediction -> correct please;

Line 283 -> "are selected" -> were selected -> review the English grammar to use the passive voice throughout the manuscript when describing the objectives, methods, and results obtained;

Line 294 -> Please describe how the imbalance ratio (table 1 data) was calculated;

Line 306 -> "Reference" -> reference;

Line 458 -> Algorithm 1 -> Algorithm 1 table should be presented within the Materials and Methods section, on line 469, before Results section;

Results

Line 470 -> Item 3.1 presented in the Results section is a description of Materials and Methods and should be presented in the correct section (Materials and Methods) and not in the Results section;

Line 503-518 -> This paragraph is Materials and Methods description, should not be at Results section also, please review;

There are several sections of the results section that are "discussion". Please revise so that the Results section contains only a description of the results obtained. Or else present both sections as "Results and Discussion".

Discussion

This is not a discussion, but rather a conclusions section, which is very vague, divergent and long. Please revise, change the name of the section to Conclusions and describe what your results reveal as I move forward in the field.

Author Response

We sincerely thank the reviewer for the meticulous attention to grammatical details and formatting issues throughout the manuscript. We are also grateful for your valuable guidance on the appropriate placement and organization of various sections, which has greatly improved the overall structure and readability of our work.

Comments 1: The introduction should be revised to make it more concise and objective. Review the need to present figures 1 and 2 in the introduction as results of a methodological pre-analysis. Results not previously described or new approaches should be presented in the results section;

Response 1: Thank you for your valuable suggestion. In accordance with your recommendation, we have removed the methodological pre-analysis content and the associated Figures 1 and 2 (now Figures 2 and 3, respectively) from the Introduction. This content has now been appropriately relocated to the Results and Discussion section under the newly added subsection titled “Structural and biological pre-analysis for applying GNNs on PPI Networks”. This revision ensures a clearer separation between introductory context and analytical results, and improves the overall structure and objectivity of the manuscript. These changes can be found in Section 3.1 on page 12-14, lines 485-528.

Comments 2: Line 120 -> Add spacing after "networks";

Response 2: Thank you for pointing this out. The formatting issue has been corrected in the revised manuscript and can be found on line 120.

Comments 3: Line 152 -> "We employ" -> We have employed -> review English;

Response 3: Thank you for pointing this out. The formatting issue has been corrected in the revised manuscript and can be found on line 486.

Comments 4: Line 161 -> "are generated" -> were generated -> review English;

Response 4: Thank you for pointing this out. The formatting issue has been corrected in the revised manuscript and can be found on line 494.

Comments 5: Line 162 -> "are calculated and subject to" -> were calculated and subjected to -> review English;

Response 5: Thank you for pointing this out. The formatting issue has been corrected in the revised manuscript and can be found on line 502.

Comments 6: Line 246-258 -> "The key contributions can be summarized..." -> this section should be revised to include the objectives of the work; it is suggested that "contributions" be moved to the conclusions section.

Response 6: Thank you for the valuable suggestion. As advised, we have revised this section to clearly state the objectives of the work. The original “key contributions” content has been moved to the Conclusion section, and the updated version now reflects the aims of our study more appropriately. These changes can be found in Section 1 on page 5, lines 206-213.

Comments 7: Line 260 -> GC-VF -> Introduce GC-VF abbreviation in the main text and in the caption of figure 3, as it was only introduced in the abstract;

Response 7: Thank you for pointing this out. We have now introduced the abbreviation “GC-VF” at its first occurrence in the main text (Line 215) and added the full name to the caption of Figure 1 (previously Figure 3, Line 256) to ensure clarity and consistency throughout the manuscript.

Comments 8: Figure 3 -> Legend -> "Virulence factor predication" -> Virulence factor prediction -> correct please;

Thank you for pointing this out. We have corrected the typographical error in the legend by replacing “Virulence factor predication” with “Virulence factor prediction.” The revision can be found in Figure 1 (located below line 255).

Comments 9: Line 283 -> "are selected" -> were selected -> review the English grammar to use the passive voice throughout the manuscript when describing the objectives, methods, and results obtained;

Response 9: Thank you for your comment. We have reviewed the manuscript thoroughly and revised the verb tenses to ensure consistent and appropriate use of passive voice, especially when describing procedures, experimental settings, and results.

Comments 10: Line 294 -> Please describe how the imbalance ratio (table 1 data) was calculated;

Response 10: Thank you for your comment. We have added a description of how the imbalance ratio was calculated in the revised manuscript (Page 6, Lines 249–253). Specifically, the imbalance ratio is defined as the number of samples in the majority class divided by that in the minority class, which is applicable in binary classification settings.

Comments 11: Line 306 -> "Reference" -> reference

Response 11: Thank you for pointing this out. The formatting issue has been corrected in the revised manuscript and can be found on line 271.

Comments 12: Line 458 -> Algorithm 1 -> Algorithm 1 table should be presented within the Materials and Methods section, on line 469, before Results section;

Response 12: Thank you for pointing this out. The formatting issue has been corrected in the revised manuscript and can be found on line 436, before Results section.

Comments 13: Line 470 -> Item 3.1 presented in the Results section is a description of Materials and Methods and should be presented in the correct section (Materials and Methods) and not in the Results section;

Response 13: Thank you for pointing this out. In accordance with your suggestion, we have moved the content originally presented in Section 3.1 of the Results to the Materials and Methods section (now Section 2.5, pages 11–12) in the new version, to ensure appropriate manuscript structure and logical flow.

Comments 14: Line 503-518 -> This paragraph is Materials and Methods description, should not be at Results section also, please review;

Thank you for your careful review and helpful suggestion. Following your comment, we have relocated the paragraph originally found in the Results section (previously in lines 503–518) to the Materials and Methods section. It now appears in the revised subsection titled Experimental Settings and Evaluation Metrics (Section 2.6, pages 12–13, lines 469-482). Additionally, we have reorganized the baseline descriptions into a bullet-point format to improve readability and presentation clarity.

Comments 15: There are several sections of the results section that are "discussion". Please revise so that the Results section contains only a description of the results obtained. Or else present both sections as "Results and Discussion".

Response 15: Thank you for your suggestion. We agree with your comment and have revised the manuscript accordingly. The original Results section has been renamed to "Results and Discussion" to appropriately reflect the inclusion of interpretative content alongside the presentation of results.

Comments 16: This is not a discussion, but rather a conclusions section, which is very vague, divergent and long. Please revise, change the name of the section to Conclusions and describe what your results reveal as I move forward in the field.

Response 16: Thank you for your suggestion. In accordance with your comment, we have renamed the section to Conclusions and revised its content to improve focus and clarity. Specifically, we have removed overly general and divergent statements, and restructured the section to clearly summarize our key contributions and experimental findings. The revised conclusion emphasizes how our results advance the field by shifting VF prediction from sequence-based to network-based modeling, and outlines the broader implications for future research in computational pathogen biology.

We sincerely thank the reviewer for the meticulous and thoughtful evaluation of our manuscript. Your comments have been invaluable in identifying critical issues related to the structure of the paper, such as the inappropriate placement of results in the Introduction and the mixing of Materials and Methods with the Results section. You also provided detailed suggestions on grammar, tense consistency, and terminology usage. These are areas we had previously overlooked, and your insights have greatly contributed to improving the clarity, coherence, and overall rigor of our work. The revision process has not only enhanced the quality of the manuscript but also deepened our understanding of scientific writing. We are truly grateful for the time and effort you devoted to helping us improve this work.

Reviewer 2 Report

Comments and Suggestions for Authors

Summary
The authors introduce GC-VF, a graph-based framework that turns virulence factor prediction into an imbalanced node-classification task on protein–protein interaction networks. They encode protein sequences into embeddings, build local subgraphs around each protein, and then apply two self-supervised modules: a generative attribute-reconstruction autoencoder and a node-level contrastive learner. Node representations feed into a classifier trained with focal loss to handle the minority VF class. Experiments on three bacterial PPI datasets show GC-VF outperforming BLAST, VirulentPred 2.0, DeepVF and DTVF across AUPRC, AUROC, sensitivity and F1-score. Extensive ablation studies and hyperparameter sweeps demonstrate the value of each component and guide optimal settings.

Strengths

The paper clearly motivates the use of PPI topology and protein features simultaneously, filling a gap in VF identification.

Combining generative reconstruction with local-local contrast at the node level is an elegant solution to the severe class imbalance in real PPI graphs.

Evaluation on three distinct, naturally imbalanced bacterial networks adds credibility and highlights robustness.

Thorough ablation confirms that both self-supervised modules, edge weights and focal loss each contribute materially to performance.

Hyperparameter sensitivity analyses for noise, temperature, subgraph size and loss weights are comprehensive and offer practical guidance.

Recommendations

The protein sequence encoder relies on a pre-trained Skip-Gram + one-hot scheme and a four-layer Conv/BiGRU network. Please specify how that model was pre-trained, the corpora used, and why a 256-dim embedding was chosen.

In the contrastive module, the negative-sampling strategy mixes intra- and inter-view negatives but omits exact sampling counts. Report the number of negatives per anchor and whether hard-negative mining was considered.

Baselines focus on sequence-based and hybrid models; consider including an existing PPI-based GNN (e.g., GraphProt) to show gains truly stem from the self-supervised design rather than simply using a graph.

The homophily analysis and random-graph comparison are interesting but under-specified. Detail the randomization algorithm and confirm that class proportions and degree distributions were preserved.

Training and inference runtimes are not mentioned.

Conclusion
GC-VF offers a compelling, well-engineered framework that leverages PPI networks and self-supervision to tackle imbalanced VF identification. Clarifying encoder pretraining, detailing contrastive sampling, reporting variability, expanding baselines and generalization tests, and sharpening methodological transparency will make the manuscript strong.

Author Response

We sincerely thank the reviewer for the thorough and constructive feedback, and for recognizing the strengths of our work. Your detailed recommendations and insightful suggestions have significantly improved the quality and clarity of our manuscript.

Comments 1: The protein sequence encoder relies on a pre-trained Skip-Gram + one-hot scheme and a four-layer Conv/BiGRU network. Please specify how that model was pre-trained, the corpora used, and why a 256-dim embedding was chosen.

Response 1: We directly adopted the amino acid embeddings pre-trained in reference [41]. These embeddings were obtained using the Skip-Gram model trained on the SHS148k dataset from the STRING database, with a context window size of 7 and a negative sampling size of 5, aiming to capture co-occurrence similarities between amino acids. Reference [44] also utilized the pre-trained embeddings from [41] and employed the Protein Feature Encoding module to generate protein representations, achieving good predictive performance on PPI datasets. Based on its demonstrated effectiveness in related tasks, we adopted the same design for protein representation construction and used the 256-dimensional protein representations generated by the Protein Feature Encoding module as input features of GNN. The corresponding details have been added in Section 2.2 on page 6-7, lines 263-277.

Comments 2: In the contrastive module, the negative-sampling strategy mixes intra- and inter-view negatives but omits exact sampling counts. Report the number of negatives per anchor and whether hard-negative mining was considered.

Response 2:

In our contrastive learning framework, we sample PPI subgraphs starting from each target node as the original view and apply Gaussian noise augmentation to create the enhanced view. For intra-view contrastive learning, we compare the target node with all other nodes within the same original subgraph view (local subgraph view), using all remaining nodes (excluding the target node itself) as negatives. For inter-view contrastive learning, we compare the target node from the original view with all other nodes in the Gaussian noise-augmented view, again excluding the corresponding target node position in the enhanced view. In both cases, the number of negatives per anchor is (subgraph-size - 1). These changes can be found in Section 2.3.3 on page 10, line 399.

Hard-negative mining was not implemented in our current work, as we focused on establishing the effectiveness of the dual-view contrastive framework with standard sampling. We appreciate the valuable suggestion, as incorporating hard negatives can indeed enhance the contrastive signal, especially in biologically complex networks like PPI. As part of future work, we plan to explore hard-negative mining strategies, such as selecting negatives based on embedding similarity or nodes with similar topological structures but different functional roles, which may further improve the discriminative power of the learned representations.

Comments 3: Baselines focus on sequence-based and hybrid models; consider including an existing PPI-based GNN (e.g., GraphProt) to show gains truly stem from the self-supervised design rather than simply using a graph.

Response 3: We appreciate this insightful suggestion. Although we did not include existing PPI-based GNN baselines such as GraphProt in our main comparison, we have partially addressed this concern through the ablation studies presented in Section 3.5 on page 18-19. Specifically, the variant GC-VF w/o SL removes both the generative and contrastive components, retaining only the supervised GNN. As shown in Table 3, this variant consistently underperforms compared to the full GC-VF model, indicating that the observed performance gains primarily stem from the self-supervised learning design rather than the use of graph structures alone.

Comments 4: The homophily analysis and random-graph comparison are interesting but under-specified. Detail the randomization algorithm and confirm that class proportions and degree distributions were preserved.

Response 4:

We sincerely thank the reviewer for pointing out this important detail and for recognizing the homophily analysis and random-graph comparison as interesting.

In our initial analysis, we generated random graphs that preserved the same number of nodes, edges, and class distribution proportions as the original PPI networks. However, we did not consider preserving the degree distribution. Following your valuable suggestion, we have improved our random graph generation method by adopting the Configuration Model to preserve the degree distribution, and we have updated the comparison results in Figure 2 and added the corresponding methodological details. These changes can be found in Section 3.1 on page 12, lines 494–503. This randomization algorithm represents each node's degree as a corresponding number of "stubs" (half-edges), then randomly pairs all stubs to form complete edges. This approach maintains the global degree distribution while simulating an average random state where edge formation is independent of node classes. This allows us to isolate the impact of class imbalance on homophily and demonstrates that the observed high homophily in PPI networks stems from genuine biological properties rather than statistical artifacts of class distribution. The consistently higher homophily in real PPI networks compared to these randomized graphs confirms that the homophilic tendency in PPI networks is an intrinsic biological characteristic, making PPI networks particularly well-suited for GNN-based methods that rely on the "birds of a feather flock together" principle.

Comments 5: Training and inference runtimes are not mentioned.

Response 5: Thank you for this important observation. We have now included the training and inference runtime information in the revised manuscript. Specifically, we added Table 4 and corresponding content which reports the runtime measurements averaged across multiple independent runs using optimal hyperparameters for each dataset. These changes can be found in Section 3.6 on page 20, lines 693–703. The results show that both training and inference times correlate positively with graph scale, as expected.

First and foremost, we sincerely thank the reviewer for recognizing the value of our work, which means a great deal to us. Your insightful comments not only reflect a deep understanding of the field but also prompted us to carefully reconsider and reflect on our research. In response to your suggestions, we have thoroughly revised and supplemented the manuscript, further improving its content and structure as well as enhancing the quality and clarity of our work. We truly appreciate your patient and meticulous review and constructive feedback, which have contributed significantly to the scientific rigor and robustness of this study and provided valuable guidance for our future research.

Once again, we are grateful for the time and effort you have devoted to reviewing this manuscript, and we look forward to continuing to improve and expand our research under your guidance and support.